# The Relationship between Intracranial Pressure and Visual Field Zones in Normal-Tension Glaucoma Patients

**DOI:** 10.3390/diagnostics13020174

**Published:** 2023-01-04

**Authors:** Akvile Stoskuviene, Lina Siaudvytyte, Ingrida Januleviciene, Antanas Vaitkus, Evelina Simiene, Viktorija Bakstyte, Arminas Ragauskas, Gal Antman, Brent Siesky, Alon Harris

**Affiliations:** 1Eye Clinic, Lithuanian University of Health Sciences, Eiveniu Str. 2, 50161 Kaunas, Lithuania; 2Neurology Clinic, Lithuanian University of Health Sciences, Eiveniu Str. 2, 50161 Kaunas, Lithuania; 3Health Telematics Science Centre of Kaunas University of Technology, Studentu Str. 50, 51368 Kaunas, Lithuania; 4Department of Ophthalmology, Rabin Medical Center, Petah Tikva 49100, Israel; 5Department of Ophthalmology, Icahn School of Medicine at Mount Sinai, New York, NY 10029, USA

**Keywords:** normal-tension glaucoma, intracranial pressure, translaminar pressure difference, visual field zones

## Abstract

Growing evidence suggests that intracranial pressure (ICP) plays an important role in the pathophysiology of glaucoma, especially in normal-tension glaucoma (NTG) patients. Controversial results exist about ICP’s relationship to visual field (VF) changes. With the aim to assess the relationship between ICP and VF zones in NTG patients, 80 NTG patients (age 59.5 (11.6) years) with early-stage glaucoma were included in this prospective study. Intraocular pressure (IOP) (Goldmann), visual perimetry (Humphrey) and non-invasive ICP (via a two-depth Transcranial Doppler, Vittamed UAB, Lithuania) were evaluated. Translaminar pressure difference (TPD) was calculated according to the formula TPD = IOP − ICP. The VFs of each patient were divided into five zones: nasal, temporal, peripheral, central, and paracentral. The average pattern deviation (PD) scores were calculated in each zone. The level of significance *p* < 0.05 was considered significant. NTG patients had a mean ICP of 8.5 (2.4) mmHg. Higher TPD was related with lower mean deviation (MD) (*p* = 0.01) and higher pattern standard deviation (PSD) (*p* = 0.01). ICP was significantly associated with the lowest averaged PD scores in the nasal VF zone (*p* < 0.001). There were no significant correlations between ICP and other VF zones with the most negative mean PD value. (*p* > 0.05). Further studies are needed to analyze the involvement of ICP in NTG management.

## 1. Introduction

Population-based studies demonstrated that approximately 50% of patients with primary open-angle glaucoma (POAG) have an IOP level constantly within normal ranges, and are diagnosed with normal-tension glaucoma (NTG) [1]. Despite the significant prevalence of NTG, the mechanisms underlying the pathophysiology of the disease remain unclear. Studies have demonstrated many additional controllable risk factors for POAG including lower ocular perfusion pressure (OPP), reduced ocular blood flow, and lower arterial blood pressure (BP). As glaucoma is multifactorial, all of these factors may play an important role in NTG pathophysiology [2,3].

Recently, researchers have focused on intracranial pressure (ICP) and translaminar pressure difference (TPD = IOP − ICP) as a component having a potential role in glaucomatous optic neuropathy [4,5,6,7]. The optic nerve is unique as it is affected both by IOP within the eye and also by ICP, as it is surrounded by cerebrospinal fluid (CSF) in the subarachnoid space. The difference in pressure between these two zones may lead to the injury of ganglion cell axons that transverse the lamina cribrosa [8,9,10]. Several studies reported reduced ICP in POAG patients, particularly in NTG [5,6,11,12,13,14]. Experimental studies revealed the influence of ICP on optic nerve structural changes, similar to glaucomatous optic neuropathy [8,9,15]. Although some small studies have reported that ICP is not reduced in glaucoma [16,17], the literature overall largely supports the influence of reduced ICP in the development and progression of glaucoma. Controversial results exist about ICP’s relationship to NTG and VF changes [4,18,19]. Thus, the aim of our study was to assess the relationship between ICP and VF zones.

## 2. Materials and Methods

Eighty patients (Caucasians) with early stage NTG were included in a prospective study.

The study was approved by Kaunas Regional Biomedical Research Ethics Committee and performed according to the tenets of the Declaration of Helsinki, with patients signing informed consent. Three hundred NTG patients referred to the Eye Clinic, Lithuanian University of Health Sciences were examined between January and October 2018. All subjects underwent a complete ophthalmological and neurological examination and 80 patients met the inclusion criteria.

The inclusion criteria were: clinical diagnosis of NTG confirmed by glaucoma specialist (characteristic optic nerve head changes, optic nerve changes and nerve fiber layer loss using Heidelberg Retina Tomograph (HRT), glaucomatous VF changes, an IOP of less than 21 mmHg before treatment). Only patients with early stage glaucomatous VF defects according Hoddap–Parrish–Anderson criteria [20] were included in the study. All examinations were performed on one eye, which was chosen randomly.

All patients were examined by a neurologist to exclude neurological disorders that could affect ICP (such as pseudotumor cerebri, intracranial tumors, any cranial surgery), usage of oral medications, including carbonic anhydrase inhibitors due to their known effects on ICP.

Additionally, patients with uncontrolled systemic diseases, pregnant or nursing women, and patients with a history of allergy to local anesthetics, orbital/ocular trauma or other diseases that could bias the study results were excluded from the study. Current medical treatment, including topical IOP-lowering drugs, was continued.

The following measurements were recorded during the study: body mass index (BMI, calculated as body mass divided by the square of height), best-corrected visual acuity (BCVA, using Snellen chart), Goldmann applanation tonometry, perimetry (Humphrey 24-2, Sita Fast, Zeiss, Dublin, CA, USA), confocal laser scanning tomography for optic nerve structural parameters (HRT, Heidelberg Retina Tomograph, Heidelberg Engineering, Heidelberg, Germany), non-invasive ICP (Vittamed UAB, Kaunas, Lithuania), and TPD (calculated as the difference between IOP and ICP), BP was measured with a digital automatic blood pressure monitor (Omron M6 Comfort, HEM-7360-E; Omron Healthcare, Kyoto, Japan). Patients were seated and BP measurements were taken after 5 min of rest. Measurements were taken twice within intervals of 5 min. A third measurement was taken only if there was a difference in systolic blood pressure (SBP) more than 10 mm Hg or diastolic blood pressure (DBP) more than 5 mmHg [21]. The BP of the patient was calculated as the mean between the two closest readings. Mean ocular perfusion pressure then was calculated according to the formula (MOPP) = 2/3(mean arterial pressure−IOP), where mean arterial pressure (MAP) = DBP + 1/3(SBP−DBP). HRT results are outside the scope of this manuscript and are reported separately.

Non-invasive ICP was measured using a two-depth Transcranial Doppler (TCD) with patients lying in a supine position and a head frame with a fixed ultrasound transducer for placement over the closed eyelid and a duration of the measurement of approximately 10 minutes. The details of this technique are described in our previous article [22].

The VFs of each patient were divided into five zones: nasal, temporal, peripheral, central and paracentral (Figure 1). The average pattern deviation (PD) scores were calculated in each zone [23]. PD values were selected as they are assumed to remove the diffuse field loss due to cataracts [24]. Two VF tests were performed and the average scores of both tests were calculated.

Statistical analysis was performed using the computer program SPSS 23.0 for Windows. Methods of descriptive statistics defined all variables. The analysis of the quantitative variables included the calculation of the mean and standard deviation (x(SD)). Association between categorical variables or continuous variables was assessed by Spearman’s or Pearson’s correlation. Associations between VF zones with the most negative averaged PD scores and ICP, IOP, BP, age, and BMI were tested in multivariate analysis. Multifactorial linear regression models were applied to evaluate ICP and TPD associations with IOP and BP by adjusting for age and BMI.

The level of significance *p* < 0.05 was considered significant.

## 3. Results

Eighty NTG patients (24 % males) with a mean age of 59.5 (11.6) years were included in this prospective study. The patient’s characteristics are provided in Table 1.

Associations between ICP, TPD and systemic and functional VF parameters are shown in Table 2 and Table 3. Lower ICP was correlated with lower IOP (*p* < 0.001) and lower systolic BP (*p* = 0.02). Higher TPD was related with lower mean deviation (MD) (*p* = 0.01) and higher PSD (*p* = 0.01). ICP and TPD were significantly associated with the most negative averaged PD scores in the nasal VF zone (*p* = 0.001). There were no significant correlations between ICP or TPD and other VF zones with the lowest averaged PD value within them (*p* > 0.05).

The lowest averaged PD value within the nasal VF zone was also correlated with diastolic BP (*p* = 0.02). The lowest averaged PD value in the central VF zone was associated with diastolic BP (*p* = 0.02), systolic BP (*p* = 0.01) and OPP (*p* = 0.02). The lowest averaged PD scores within the paracentral VF zone were correlated with lower BMI (*p* = 0.006).

Associations between VF zones with the most negative mean PD values and systemic parameters remained statistically significant in multivariate analysis. The lowest averaged PD value within the nasal VF zone correlated with ICP (beta 0.40, *p* < 0.001) and IOP (beta −0.26, *p* = 0.03); and no statistical significance (*p* > 0.05) was found with age, BMI, or BP. The most negative averaged PD scores in paracentral VF zone significantly correlated with BMI (beta 0.06, *p* = 0.04) and was not statistically significantly (*p* > 0.05) correlated with age, IOP, ICP and BP. There were no associations between the most negative averaged PD scores within the temporal VF zone and other parameters (*p* > 0.05), the most negative averaged PD scores within the peripheral VF zone and other parameters (p>0.05), the most negative averaged PD scores within central VF zone and other parameters (p>0.05) in multivariate analysis.

In age-adjusted and BMI–adjusted multivariate linear regression analysis, both ICP and TPD remained significantly (*p* < 0.001) associated with IOP and BP, and was not statistically significantly associated with age and BMI (*p* > 0.05). Results are presented in Table 4.

## 4. Discussion

Using non-invasive methods, our study identified a mean ICP value of 8.5 (2.4) mmHg in NTG patients. We also found several statistically significant correlations between ICP, TPD and VF parameters. Specifically, higher TPD significantly correlated with lower MD and higher PSD. In addition, ICP and TPD were significantly correlated with the most negative averaged PD scores within the nasal VF zone only as compared to other VF zones.

The two-depth Transcranial Doppler mean ICP value in our study was similar to the reported mean ICP as measured by both lumbar puncture [5,18] and the same non-invasive device in our previous studies [13,22]. However, Linden et al. study results contradict most previous reports and reviews on ICP in glaucoma patients as they did not find lower ICP in NTG patients. They offered several ideas to explain this discrepancy, including differences in assessments of IOP in an upright posture and ICP in horizontal posture [16]. Importantly, even though the ICP and IOP were measured simultaneously in their study, CSF pressure measured by lumbar puncture is not the same as ICP in an upright posture. CSF pressures in the spinal canal and cranial cavity differ in upright posture due to the biophysical characteristics of the CSF system, i.e., distensibility of the spinal dura [25]. A retrospective study of homogenous Caucasians with NTG also did not find lower lumbar CSF pressure [19], in contrast to previous retrospective and prospective studies using invasive lumbar CSF pressure measurement [5,18].

In the study by Loiselle et al., ICP was evaluated non-invasively as the response of the inner ear to two tones at specified levels and frequencies, named as distortion product otoacoustic emission (DPOAE). They assessed the relationship between DPOAE phase and body position and also compared it between POAG, NTG and control groups, and reported no reduced ICP in glaucoma patients [17]. Even though studies performed by Linden et Loiselle reported no significant ICP value on NTG patients, they both demonstrated methodological issues and their conclusions could not be paralleled with a number of well-powered studies [26].

Some previous studies reported that ICP had a tendency to decrease by age [27,28]. We did not find correlations between ICP or TPD and age, similarly to Ren et al. or Pircher et al. [19,29]. Also, we did not find correlations between ICP or TPD and BMI, similarly to results published by Bono and colleagues [30], whereas Berdhal et al. reported a linear relationship between CSF pressure and BMI while IOP was unaffected by BMI and CSF pressure. Authors hypothesized a possible protective impact of increasing BMI for glaucoma; if IOP considered constant, the TPD should decrease with increasing BMI [31]. Ren et al. also reported a tendency that CSF pressure increases and IOP remains stable with increasing BMI [29].

Previous clinical and experimental studies have also reported existing associations between CSF pressure, BP, IOP and BMI [4]. Ren et al. reported that CSFP, IOP, and BP were correlated with each other and proposed there could be a systemic mechanism simultaneously adjusting all three of them [29]. Similarly, in our study, lower ICP was significantly correlated with lower IOP and lower systolic BP, whereas higher TPD was associated with higher IOP. Accepting age and BMI as potential confounders, linear regression analysis was conducted. ICP and TPD remained significantly correlated with lower IOP and lower BP after age and BMI adjustment.

In this study, we found several statistically significant correlations between ICP, TPD and VF parameters. Higher TPD was correlated with lower MD and higher PSD. Therefore, higher TPD could be estimated as a risk factor playing a role in NTG pathogenesis and in VF changes. Similarly, Ren et al. found that the extent of glaucomatous VF loss was negatively correlated with the height of the CSF pressure and positively correlated with TPD [18]. However, others did not find a correlation between TPD and VF changes [19] or between ICP and VF changes [5,16].

Therefore, we found that ICP and TPD were significantly correlated with the most negative averaged PD scores within the nasal VF zone only as compared to other VF zones. The lowest mean PD value within the nasal VF zone remained significantly associated with ICP in a multivariate analysis model. Elevated IOP was proposed as a possible risk factor for the initial nasal step in VF by Park et al. [32]. Our results support the hypothesis that in addition to elevated IOP, a decreased ICP within normal IOP range might also lead to earlier and deeper defects in the nasal VF zone. These findings are also in line with previous results suggesting decreased ICP, with or without significantly elevated IOP, may lead to glaucomatous damage [5,22].

As elevated IOP was reported as a possible risk factor for the initial nasal step in VF, the initial parafoveal VF defect had a stronger association with IOP-independent risk factors, such as hypotension or sleep apnea [32]. Ahrlich and colleagues suggested that VF changes in the central zone may be influenced by disturbed vascular autoregulation in NTG patients, as disturbances of vascular autoregulation may predispose unstable OPP with changes in BP or IOP [33]. Interestingly, systemic hypertension in its early stages may have a relatively protecting effect on the central VF zone. Even though there might be enhanced blood flow to the eye in the initial stage of hypertension, the smaller arterioles become affected and cause disturbances in perfusion later with the progression of the disease. That may lead to peripheral VF defects occurring earlier than central ones [34]. Interestingly, in our study the lowest averaged PD values within the nasal VF zone correlated with higher diastolic BP while the central VF zone with the most negative mean PD value was related with lower diastolic and systolic BP and lower OPP. However, subsequently conducted multivariate analysis revealed no significant associations between these VF zones and considered systemic parameters.

We also found that the paracentral VF zone with the most negative PD value significantly correlated with lower BMI and the result remained statistically significant after multivariate analysis. The relationship between BMI, as an IOP-independent risk factor, and glaucoma remains controversial [35]. Lower BMI was related to structural optic nerve disc changes in glaucoma patients by Zheng and colleagues [35]. Choi et al. revealed that lower BMI was significantly related to faster VF progression in NTG patients [36].

Our study has several limitations to acknowledge. In our approach we utilized a non-invasive ICP two-depth TCD device measurement methodology, which may not reflect the same values as the invasive gold standard lumber puncture may produce. In addition, in our study ICP was measured in the supine position due to ICP measurement technique restrictions, while IOP was assessed in the sitting position. The difference in posture during measurement may influence biomarkers, as IOP and ICP are pulsatile parameters that are influenced by body position and the cardiac cycle. In addition, visual acuity testing was performed using Snellen charts for statistical analysis while other approaches including use of logMAR may result in different results, especially within a small sample.

Some issues evaluating VF changes should be mentioned as well. The nasal VF zone contains the two most eccentric points. A nasal step, a sign in glaucoma, also can be noticed in normal VF, other pathological conditions, as a late sign of the distal edge of the arcuate scotoma, or as an artifact [37]. We included patients with the diagnosis of early-stage NTG confirmed by an ophthalmologist. The possibility of artifacts remains but should be minimized, as changes in VF should appear glaucomatous and correspond with other clinical findings for glaucoma diagnosis [38]. Also, two VF tests were carried out and the average of both tests was scored in our study.

VF points in contiguous zones, for example nasal and paracentral, are not fully independent, whereas the statistical tests we used assume independence. We presume this should not affect the results as we analyzed VF with early glaucomatous loss. However, the adjacent zones could be important in evaluating patients with moderate or advanced glaucoma with more diffused VF defects. Further clinical studies are needed to confirm the validity of our findings.

## 5. Conclusions

Higher translaminar pressure difference was correlated with lower mean deviation and higher pattern standard deviation. Intracranial pressure was significantly associated with the lowest averaged pattern deviation values within the nasal visual field zone. These data support additional studies to reveal the mechanistic involvement of ICP in glaucoma pathophysiology, especially as a potential biomarker for NTG.

## Figures and Tables

**Figure 1 diagnostics-13-00174-f001:**
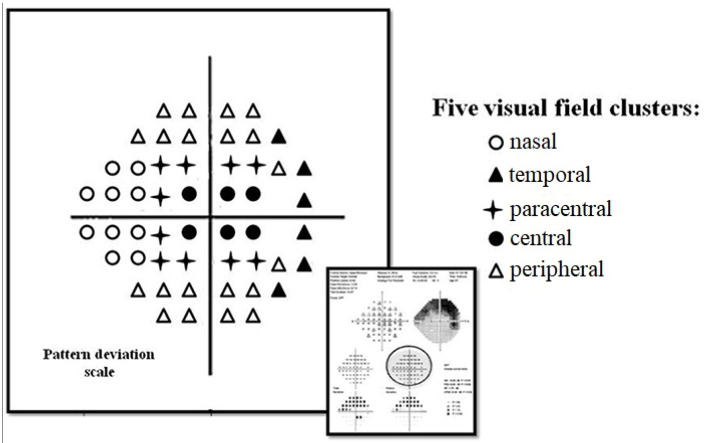
Pattern deviation scale plot of the visual field printout. Five visual field zones were defined: nasal, temporal, peripheral, central, paracentral.

**Table 1 diagnostics-13-00174-t001:** Normal-tension glaucoma patient’s characteristics.

	NTG Patients (N = 80) Mean (SD)
Male [N (%)]	19 (24%)
Mean age [years]	59.4 (11.6)
Mean body mass index [kg/m^2^]	27.1 (4.8)
Mean glaucoma treatment [years]	2.5 (3.5)
Glaucoma medications [N (%)]: β-blockers Pg analogues CAI α-agonists	
	17 (21%)
	43 (54%)
	13 (16%)
	1 (1%)
Systemic medications [N (%)]: Diuretics β-blockers ACE inhibitors Angiotensin II inhibitors Others	
	7 (9%)
	22 (28%)
	25 (31%)
	2 (3%)
	35 (44%)
Systemic diseases [N (%)]: Low arterial BP High arterial BP Diabetes mellitus Thyroid pathology Migraine	
	15 (19%)
	40 (50%)
	4 (5%)
	8 (10%)
	4 (5%)
Ophthalmological examination [mean] BCVA ICP [mmHg] IOP [mmHG] TPD [mmHg] MOPP [mmHg] Pulse [bpm] Systolic BP [mmHg] Diastolic BP [mmHg]	
	0.97 (0.1)
	8.5 (2.4)
	15.0 (2.3)
	6.3 (2.5)
	57.1 (7.6)
	68.1 (8.6)
	135.5 (16.4)
	82.9 (9.9)
VF parameters [mean] MD, dB PSD, dB VFI, dB	
	−1.8 (1.8)
	2.4 (1.7)
	97 (3)
Averaged PD scores within VF zones [mean] Nasal VF zone, dB Temporal VF zone, dB Peripheral VF zone, dB Central VF zone, dB Paracentral VF zone, dB	
	−2.5 (2.1)
	−1.9 (2.0)
	−2.2 (1.9)
	−1.2 (0.7)
	−1.8 (1.0)

ACE—angiotensin converting enzyme, BCVA—best corrected visual acuity, BP—blood pressure, CAI—carbonic anhydrase inhibitors, CDR—cup-disk ratio, HRT—Heidelberg Retina Tomography, ICP—intracranial pressure, IOP—intraocular pressure, MD—mean deviation, N—number, NTG—normal-tension glaucoma, OND—optic nerve disc, OPP—ocular perfusion pressure, PD—pattern deviation, Pg—prostaglandins, PSD—pattern standard deviation, SD—standard deviation, TPD—translaminar pressure difference, VF—visual field, VFI—visual field index.

**Table 2 diagnostics-13-00174-t002:** Associations between intracranial pressure, translaminar pressure difference and systemic parameters in normal-tension glaucoma patients.

		Age [years]	BMI [kg/m^2^]	Systolic BP [mmHg]	Diastolic BP [mmHg]	OPP [mmHg]	IOP [mmHg]
ICP [mmHg]	r	0.12	0.07	0.26	−0.03	0.05	0.39
	p	(0.26)	(0.56)	(0.02) *	(0.82)	(0.65)	(<0.001) *
TPD [mmHg]	r	−0.14	−0.14	−0.20	0.03	−0.16	0.41
	p	(0.21)	(0.21)	(0.07)	(0.78)	(0.16)	(<0.001) *

Significance level *p* < 0.05, * Pearson’s correlation, BMI—body mass index, BP—blood pressure, ICP—intracranial pressure, IOP—intraocular pressure, OPP—ocular perfusion pressure, TPD—translaminar pressure difference.

**Table 3 diagnostics-13-00174-t003:** Associations between functional visual field parameters and intracranial pressure, translaminar pressure difference, intraocular pressure and systemic parameters.

			ICP [mmHg]	TPD [mmHg]	IOP [mmHg]	Age [years]	BMI [kg/m^2^]	Systolic BP [mmHg]	Diastolic BP [mmHg]	OPP [mmHg]
VF zones	Nasal [dB]	r	0.36	−0.38	−0.03	−0.05	−0.10	−0.10	−0.25	−0.19
		p	(0.001) **	(0.001) **	(0.80)	(0.70)	(0.39)	(0.36)	(0.02) **	(0.09)
	Temporal [dB]	r	0.03	−0.14	−0.20	−0.08	−0.03	−0.02	−0.18	0.02
		p	(0.79)	(0.23)	(0.07)	(0.49)	(0.81)	(0.85)	(0.87)	(0.85)
	Central [dB]	r	0.003	0.12	0.16	0.07	0.15	0.29	0.27	0.26
		p	(0.98)	(0.28)	(0.17)	(0.52)	(0.19)	(0.01) *	(0.02) *	(0.02) *
	Paracentral [dB]	r	0.04	0.05	0.10	−0.09	0.31	0.10	0.15	0.13
		p	(0.75)	(0.66)	(0.38)	(0.38)	(0.006) **	(0.39)	(0.19)	(0.26)
	Peripheral [dB]	r	0.15	−0.71	−0.07	−0.11	−0.02	−0.12	−0.13	−0.12
		p	(0.20)	(0.13)	(0.56)	(0.34)	(0.86)	(0.27)	(−0.24)	(0.30)
VF parameters	MD [dB]	r	0.19	−0.27	−0.07	−0.53	0.15	−0.04	−0.12	−0.07
		p	(0.10)	(0.01) *	(0.57)	(0.64)	(0.19)	(0.70)	(0.27)	(0.53)
	PSD [dB]	r	−0.21	0.28	−0.40	0.09	0.000	0.05	0.13	0.08
		p	(0.06)	(0.01) **	(0.72)	(0.42)	(1.00)	(0.63)	(0.27)	(0.48)
	VFI [%]	r	0.13	−0.19	−0.76	−0.11	−0.03	−0.03	−0.09	−0.04
		p	(0.25)	(0.10)	(0.50)	(0.35)	(0.79)	(0.82)	(0.44)	(0.72)

Significance level *p* < 0.05, * Pearson’s correlation, ** Spearman’s correlation, BMI—body mass index, BP—blood pressure, CDR—cup-disc ratio, ICP—intracranial pressure, IOP—intraocular pressure, MD—mean deviation OPP—ocular perfusion pressure, PSD—pattern standard deviation, TPD—translaminar pressure difference, VF—visual field, VFI—visual field index.

**Table 4 diagnostics-13-00174-t004:** Results of associations between ICP, TPD and potential predictors in multivariate and linear regression analysis for adjustment by age.

	Multivariate		Age Adjusted		BMI Adjusted	
Parameter	B [95% CI]	*p*-Value	B [95% CI]	*p*-Value	B [95% CI]	*p*-Value
ICP						
Age, years	−0.28 [−0.77–0.2]	0.25	–	–	−0.28 [−0.77–0.2]	0.25
BMI, kg/m^2^	0.12 [−0.94–0.12]	0.85	0.01 [−0.1–0.11]	0.9	–	–
Systolic BP, mmHg	0.11 [0.06–0.17]	<0.001	0.1 [0.05–0.14]	<0.001	0.11 [0.06–0.17]	<0.001
Diastolic BP, mmHg	−0.15 [−0.22–0.07]	<0.001	−0.13 [−0.2–−0.06]	<0.001	−0.14 [−0.21–−0.07]	<0.001
IOP, mmHg	0.42 [0.18–0.65]	0.001	0.43 [0.24–0.61]	<0.001	0.43 [0.25–0.62]	<0.001
TPD						
Age, years	0.28 [−0.02–0.08]	0.56	–	–	0.03 [−0.02–0.08]	0.08
BMI, kg/m^2^	−0.11 [−0.12–0.09]	0.11	−0.01 [–0.11–0.1]	0.9	–	–
Systolic BP, mmHg	−0.09 [−0.16–−0.03]	0.003	−0.09 [−0.14–−0.05]	<0.001	−0.11 [−0.17–−0.06]	<0.001
Diastolic BP, mmHg	0.12 [0.03–0.21]	0.007	0.13 [0.06–0.2]	<0.001	0.14 [0.07–0.21]	<0.001
IOP, mmHg	0.57 [0.39–0.75]	<0.001	0.57 [0.39–0.76]	<0.001	0.57 [0.39–0.75]	<0.001

B—Unstandardized coefficient B, CI—confidence intervals, ICP—intracranial pressure, BMI—body mass index, BP—blood pressure, IOP—intraocular pressure, OPP—ocular perfusion pressure, TPD—translaminar pressure difference.

## Data Availability

The data that support the findings of this study are available from the corresponding author (I.J.) upon reasonable request, due to restrictions of privacy.

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
