# Peer review of "The Relationship between Intracranial Pressure and Visual Field Zones in Normal-Tension Glaucoma Patients"

_diagnostics, 2023, doi:10.3390/diagnostics13020174_

Round 1
Reviewer 1 Report
The author tried to find the relationships between intracranial pressure, translaminar pressure difference and visual field zones among NTG patients. Here are my comments:
1. The author mentioned in line 59 that 80 NTG patients were got involved. But the author also mentioned that
2. The including criteria was not clear (line 66), for example, the definition of “characteristic optic nerve head changes”, the characteristics of “earlier stage glaucomatous VF changes”. Also, for complex cases, how did the author guard the precise diagnosing of NTG? By a glaucoma specialist or all by an ophthalmologist?
3. Please supply the clear standard (criteria or definition) in regarding neurological disorders which could affect ICP, in line 70.
4. In line 76, how did the author measure and record OPP? Please reorganize the paragraph (line 76-83) and clarify the measurements as well as values recording rules of BP, OPP and so on. For example, what does these values recorded? Only one time measuring or mean value of three times?
5. The author analyzed quantitative variables by mean and SD which based on a hypothesis that data were normal distributed. But the data were analyzed by non-parametric test which based on a hypothesis that data were abnormal distributed.
6. In table 1, what did the exact meaning of “glaucoma eye drops per day”?
7. In table 1, the summation of percentages in glaucoma medications, systemic medications, and systemic disease were over 100%, the author need to clarify how they deal with these data.
8. BCVA need to transform into logMAR before statistical analyzing.
9. Did the author test the collinearity of parameters before multivariate analysis?
Author Response
Reviewer: 1
- The author mentioned in line 59 that 80 NTG patients were got involved. But the author also mentioned that
- The including criteria was not clear (line 66), for example, the definition of “characteristic optic nerve head changes”, the characteristics of “earlier stage glaucomatous VF changes”. Also, for complex cases, how did the author guard the precise diagnosing of NTG? By a glaucoma specialist or all by an ophthalmologist?
Answer: Thank you for the comment and chance to clarify, we have corrected the following paragraph: “The inclusion criteria for this study were the following: clinical diagnosis of NTG confirmed by glaucoma specialist (characteristic optic nerve head changes, optic nerve changes and nerve fiber layer loss using Heidelberg Retina Tomograph (HRT), glaucomatous VF changes and an IOP of less than 21 mmHg before treatment). Only patients with early stage glaucomatous VF defects according Hoddap –Parrish –Anderson criteria [20] were included in the study. All examinations were performed on one eye, which was chosen randomly. [20 Ref. Susanna R Jr, Vessani RM. Staging glaucoma patient: why and how? Open Ophthalmol J. 2009 Sep 17;3:59-64,]”
- Please supply the clear standard (criteria or definition) in regarding neurological disorders which could affect ICP, in line 70.
Answer: Thank you for your remark, we have now detailed neurological disorders which could affect ICP: “All patients were examined by a neurologist to exclude neurological disorders that could affect ICP (such as pseudotumour cerebri, intracranial tumors, any cranial surgery), usage of oral medications, including carbonic anhydrase inhibitors due to their known effects on ICP.”
- In line 76, how did the author measure and record OPP? Please reorganize the paragraph (line 76-83) and clarify the measurements as well as values recording rules of BP, OPP and so on. For example, what does these values recorded? Only one time measuring or mean value of three times?
Answer: Thank you for your observation. The following paragraph was specified:
BP was measured with a digital automatic blood pressure monitor (Omron M6 Comfort, HEM-7360-E; Omron Healthcare, Kyoto, Japan). Patients were seated and BP measurements were taken after 5 minutes of rest. Measurements were taken twice within interval of 5 minutes. A third measurement was taken only if there was a difference in systolic blood pressure (SBP)more than 10 mmHg or diastolic blood pressure (DBP) more than 5 mmHg [Ref. Yingfeng Zheng, Tien Y. Wong, Paul Mitchell, David S. Friedman, Mingguang He, Tin Aung; Distribution of Ocular Perfusion Pressure and Its Relationship with Open-Angle Glaucoma: The Singapore Malay Eye Study. Invest. Ophthalmol. Vis. Sci. 2010;51(7):3399-3404.] . The BP of the patient was calculated as the mean between the two closest readings. Mean ocular perfusion pressure then was calculated according the formula (MOPP) = â…” (mean arterial pressure − IOP), where mean arterial pressure (MAP) = DBP + â…“(SBP − DBP).
The Table 1 was also corrected by adding MOPP instead of OPP.
- The author analyzed quantitative variables by mean and SD which based on a hypothesis that data were normal distributed. But the data were analyzed by non-parametric test which based on a hypothesis that data were abnormal distributed.
Answer: Thank you for your remark. The following sentence was written by a mistake and was removed from the manuscript: The hypothesis of equality among more than two groups was analyzed using the Kruskal-Wallis test, among two groups – Mann-Whitney test.
- In table 1, what did the exact meaning of “glaucoma eye drops per day”?
Answer: Thank you for your observation. The expression “glaucoma eye drops per day” means how many drops patients were taken per day (as some of the patients do not get any glaucoma medications). In case do not disturb the readers, this component was removed from Table 1.
- In table 1, the summation of percentages in glaucoma medications, systemic medications, and systemic disease were over 100%, the author need to clarify how they deal with these data.
Answer: Thank you for your remark. The percentages in glaucoma medications or systemic medications was over 100% because some of the patients were taken medications from several specified medications groups at the same time. The percentages of systemic diseases was over 100% because some of the patients had more than one of specified systemic disease. Thank you for the opportunity to clarify this in the revised manuscript.
- BCVA need to transform into logMAR before statistical analyzing.
Answer: Thank you for your note. However, in this particular case, literature suggests converting data from Snellen data to logMAR is not recommended as data may become misleading by suggesting the data more reliable that they actually are.[ Ref. Elliott, David B. The good (logMAR), the bad (Snellen) and the ugly (BCVA, number of letters read) of visual acuity measurement. Ophthalmic and Physiological Optics. Ophthalmic Physiol Opt.VL - 36 IS - 4 SP - 355 EP - 358 PY - 2016]
- Did the author test the collinearity of parameters before multivariate analysis?
Answer: Thank you for your remark. The collinearity of parameters were tested before a multivariate analysis.
Please see the attachment (revised manuscript with tracked changes)

Reviewer 2 Report
The authors present an interesting study exploring the relationship between ICP and VF findings in patients with Normal-Tension Glaucoma. They conducted a prospective study of 80 NTG patients and collected multiple VF tests in addition to ICP measurement using a previously described method employing a two-depth transcranial doppler device. The authors appropriately recognize this measurement of ICP as a study limitation in the discussion. Overall, despite the limitations, this is a well conducted study with compelling findings that further support the role of reduced ICP in the pathophysiology of POAG, in particular NTG.
Introduction
- I would consider revising the statement “While other studies have not found reduced ICP in glaucoma studies” as citation 16 references a study with flawed methodology as was discussed in the reply by Baartman et al.
I would suggest re-phrasing to the following: “There is a considerable amount of evidence supporting the influence of ICP on optic nerve head structural changes. Although some small studies have reported that ICP is not reduced in glaucoma, the literature overall largely supports the influence of reduced ICP in the development and progression of glaucoma.
Results
- Define the parentheses in the Table 1 where mean values are demonstrated. I suspect this value represents the SD but this should be clarified for the reader.
Discussion
- Some of the comments in the discussion are repetitive. The discussion has great content and offers compelling conclusions but the discussion as a whole can be made more concise.
- I would add additional comments regarding the limitations of the study by Linden et al as this study reported findings and presented conclusions that contradict a number of well-powered studies.
- There are multiple instances where the grammar and syntax could be improved throughout the discussion. There are a number of times where an adverb is used to start the sentence without punctuation to follow. These should be cleaned up to improve the quality of the manuscript.
Author Response
Reviewer: 2
Introduction
- I would consider revising the statement “While other studies have not found reduced ICP in glaucoma studies” as citation 16 references a study with flawed methodology as was discussed in the reply by Baartman et al.
I would suggest re-phrasing to the following: “There is a considerable amount of evidence supporting the influence of ICP on optic nerve head structural changes. Although some small studies have reported that ICP is not reduced in glaucoma, the literature overall largely supports the influence of reduced ICP in the development and progression of glaucoma.
Answer: Thank you for your comment, we agree and the Introduction was corrected:
Although some small studies have reported that ICP is not reduced in glaucoma [16,17], the literature overall largely supports the influence of reduced ICP in the development and progression of glaucoma.
Results
- Define the parentheses in the Table 1 where mean values are demonstrated. I suspect this value represents the SD but this should be clarified for the reader.
Answer: Thank you for your remark, Table 1 was supplemented per your request.
Discussion
- Some of the comments in the discussion are repetitive. The discussion has great content and offers compelling conclusions but the discussion as a whole can be made more concise.
Answer: Thank you for your observation, we fully agree and have significantly edited the entire document, especially the discussion section to be more succinct, have less overlap and improve readability in the revised manuscript.
- I would add additional comments regarding the limitations of the study by Linden et al as this study reported findings and presented conclusions that contradict a number of well-powered studies.
Answer: Thank you for your comments. We have significantly revised the discussion and the issues regarding the limitations of the study by Linden et al was added: Even though studies performed by Linden et Loiselle reported no significant ICP value on NTG patients, they both demonstrated methodological issues and their conclusions could not be paralleled with a number of well-powered studies.”
- There are multiple instances where the grammar and syntax could be improved throughout the discussion. There are a number of times where an adverb is used to start the sentence without punctuation to follow. These should be cleaned up to improve the quality of the manuscript.
Answer: Thank you for the opportunity to improve grammar and syntax of the article, we have significantly edited the article throughout in the revised manuscript.
Please see the attachment (revised manuscript with tracked changes)

Round 2
Reviewer 1 Report
Thank you for your point-to-point responses. All responses were reasonable and acceptable. The only thing I was concerned about was the calculation of visual acuity. Maybe it’s true that transferring Snellen data to logMAR data was not scientific enough during showing data, but using logMAR during calculation is still highly recommended.
Using Snellen directly when statistical analyzing would lead to results bias because for a patient with Snellen 6/6 was not means the visual acuity is ten times better than that of Snellen 20/200.
Author Response
Thank you very much for your comment, we understand the limitation and while we believe that our presentation of the data is the best approach, we agree that using Snellen for statistical analysis for visual acuity instead of logMAR may represent a limitation of our study. We therefore have edited the manuscript correspondingly acknowledging this and added discussion per your comment in the “limitations” section, thank you.
Please see the attachment.
